

# The impact of Docker containers on the performance of genomic pipelines

Paolo Di Tommaso[1,2], Emilio Palumbo[1,2], Maria Chatzou[1,2],
Pablo Prieto[1,2], Michael L. Heuer[3] and Cedric Notredame[1,2]

[1] Bioinformatics and Genomics Program, Centre for Genomic Regulation (CRG), Barcelona, Spain
[2] Universitat Pompeu Fabra (UPF), Barcelona, Spain
[3] Department of Bioinformatics Research, National Marrow Donor Program, Minneapolis, MN, United States

## ABSTRACT

Genomic pipelines consist of several pieces of third party software and, because of their experimental nature, frequent changes and updates are commonly necessary thus raising serious deployment and reproducibility issues. Docker containers are emerging as a possible solution for many of these problems, as they allow the packaging of pipelines in an isolated and self-contained manner. This makes it easy to distribute and execute pipelines in a portable manner across a wide range of computing platforms. Thus, the question that arises is to what extent the use of Docker containers might affect the performance of these pipelines. Here we address this question and conclude that Docker containers have only a minor impact on the performance of common genomic pipelines, which is negligible when the executed jobs are long in terms of computational time.

## INTRODUCTION

Genomic pipelines usually rely on a combination of several pieces of third party research software. These applications tend to be academic prototypes that are often difficult to install, configure and deploy. A program implemented in a given environment typically has many implicit dependencies on programs, libraries, and other components present within that environment. As a consequence, a computational workflow constructed in one environment has little chance of running correctly in another environment without significant effort (*Garijo et al., 2013*). In the past, machine virtualization technology was proposed as an answer to this issue (*Howe, 2012*; *Gent, 2013*). However, this approach has some significant disadvantages. Virtual machine images are large (typically several gigabytes) because they require a complete copy of the operating system (OS) files. Even to add or change just a single file, an overall copy of the virtual machine needs to be assembled and deployed. Moreover it is difficult, if not impossible, to reuse pieces of software or data inside a virtual machine. Their content tends to be opaque i.e., not systematically described or accessible with a standard tool/protocol (*Hinsen, 2014*).

Docker containers technology (http://www.docker.com) has recently received an increasing level of attention throughout the scientific community because it allows

Corresponding author
Paolo Di Tommaso,
paolo.ditommaso@crg.eu

applications to run in an isolated, self-contained package that can be efficiently distributed and executed in a portable manner across a wide range of computing platforms (*Gerlach et al., 2014*; *Boettiger, 2015*).

The most obvious advantage of this approach is to replace the tedious installation of numerous pieces of software, with complex dependencies, by simply downloading a single pre-built ready-to-run image containing all the software and the required configuration.

Another advantage of Docker is that it runs each process in an isolated container that is created starting from an immutable image. This prevents conflicts with other installed programs in the hosting environment, and guarantees that each process runs in a predictable system configuration that cannot change over time due to misconfigured software, system updates or programming errors.

A container only requires a fraction of a second to start and many instances can run in the same hosting environment. This is possible because it runs as an isolated process in userspace on the host operating system, sharing the kernel with other containers.

A study from IBM Research shows that Docker containers introduce a negligible overhead for CPU and memory performance, and that applications running in a container perform equally or better when compared to traditional virtual machine technology in all tests (*Felter et al., 2014*).

A question that arises is to what extent the use of Docker containers might affect the performance of a computational workflow when compared to "native" execution. In this work, we assess the impact of Docker containers on the performance of genomic pipelines using a realistic computational biology usage scenario based on the re-computation of selected subsets of the mouse ENCODE analysis. ENCODE, commonly defined as the Encyclopedia of DNA elements, is a large-scale genomic annotation project that aims at characterizing the transcriptome, the regulatory state and a part of the epigenome in a selected set of cell lines (*ENCODE Project Consortium, 2012*). As a proof of principle, we used our implementation to analyze randomly sampled RNA-Seq reads from two mouse embryo brain samples (CNS), at day 14 and day 18, in two bioreplicates.

## METHOD

In order to evaluate the impact of Docker usage on the execution performance of bioinformatics tools, we benchmarked three different genomic pipelines. A comparison of the execution times was made running them with and without Docker along with the same dataset. The tests were executed using a cluster node HP BL460c Gen8 with 12 cpus Intel Xeon X5670 (2.93 GHz), 96 GB of RAM and running on Scientific Linux 6.5 (kernel 2.6.32-431.29.2.el6.x86_64).

Tests were executed using Docker 1.0 configured with *device mapper* as the storage driver. Docker images used for the benchmark were built starting from a Scientific Linux 6.5 base image. The compute node was reserved for the benchmark execution (this means that no other workload was dispatched to it); moreover, to prevent any possible network latencies that could affect the execution times in an unpredictable manner, all tests were executed using the node local disk as main storage.

All three pipelines are developed with Nextflow, a tool that is designed to simplify the deployment of computational pipelines across different platforms in a reproducible manner (*Di Tommaso et al., 2014*). Nextflow provides built-in support for Docker, allowing pipeline tasks to be executed transparently in Docker containers; this allowed us to execute the same pipeline natively or run it with Docker without having to modify the pipeline code, but by simply specifying the Docker image to be used in the Nextflow configuration file.

It should be noted that when the pipeline is executed with Docker support, it does not mean that the overall pipeline execution runs "inside" a single container, but that each task spawned by the pipeline runs in its own container. The container working directory is set to a Docker volume mounted to a local directory in the hosting file system. In this way tasks can share data transparently, without affecting the flow in the pipeline execution. This approach allows a Docker based pipeline to use a different image for each different task in the computational workflow, and therefore scale seamlessly in a cluster of computers using a network shared storage such as NFS (which wouldn't be possible using the single container approach).

The overhead introduced by containers technology on the pipelines performance was estimated by comparing the mean execution time of 10 instances running with and without Docker. As the pipeline ran parallel tasks, the execution time was normalized summing up the execution time of all the tasks in each instance. The task execution time was calculated as the difference between the launch and completion system timestamps (at millisecond resolution using the *System.currentTimeMillis* Java API). Thus, it includes the container instantiation time overhead when Docker was used (but not the time needed to pull the required image which was previously downloaded).

## Benchmark 1

The first performance evaluation was carried out using a simple pipeline for RNA-Seq data analysis.

The pipeline takes raw RNA-Seq sequences as input and first maps them, by sequence alignment, to a reference genome and transcript annotation. The mapping information is then used to quantify known transcripts using the reference transcript annotation. For each processed sample, the pipeline produces as output a table of relative abundances of all transcripts in the transcript annotation.

The pipeline was run 10 times using the same dataset with and without Docker. The RNA-Seq data was taken from the ENCODE project and contained randomly sampled (10% of the original) Illumina paired-end sequences from two mouse embryo brain samples (CNS), at day 14 and day 18, in 2 bioreplicates. Each run executed 9 tasks: a first *index* task using Bowtie, then 4 *mapping* tasks using Tophat2 and finally 4 *transcript* tasks using the Cufflinks tool. Both mapping and transcript tasks were executed in parallel. The following versions of these tools were used: Samtools 0.1.18 (*Li et al., 2009*), Bowtie2 2.2.3 (*Langmead & Salzberg, 2012*), Tophat-2.0.12 (*Kim et al., 2013*), Cufflinks 2.2.1 (*Trapnell et al., 2010*).

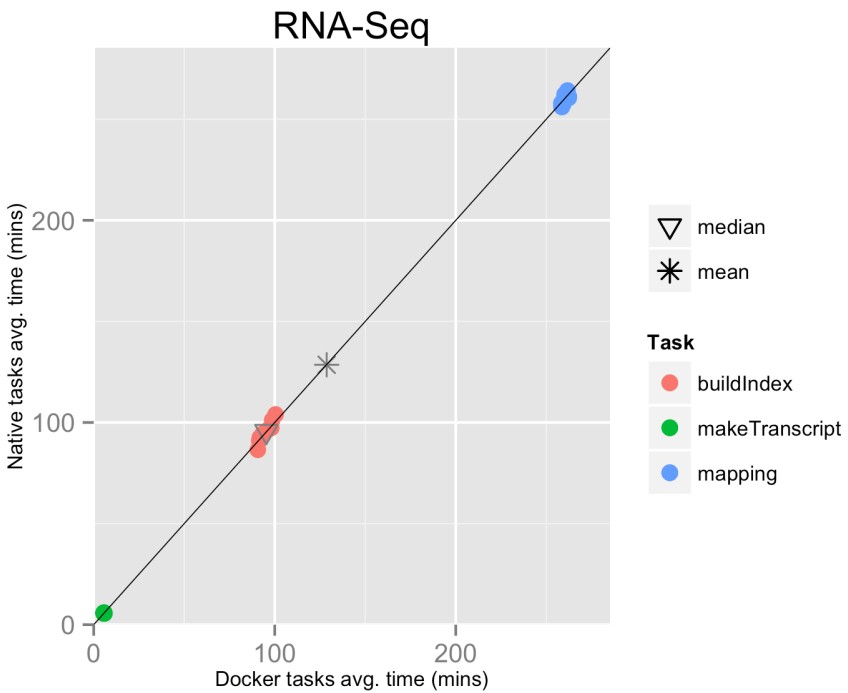

**Figure 1 RNA-Seq pipeline tasks, native vs. Docker mean execution times.** Time elapsed (in minutes) to complete since the submission including the container instantiation. Each point represents the mean task time for the same type in one pipeline run.

**Table 1 Mean execution times for pipelines and tasks with and without Docker.** Time is expressed in minutes. The mean and the standard deviation were estimated from 10 separate runs. Slowdown represents the ratio of the mean execution time with Docker to the mean execution time when Docker was not used.

| Pipeline | Tasks | Mean task time | | Mean execution time | | Execution time std. deviation | | Slowdown |
|---|---|---|---|---|---|---|---|---|
| | | Native | Docker | Native | Docker | Native | Docker | |
| RNA-Seq | 9 | 128.5 | 128.7 | 1,156.9 | 1,158.2 | 13.5 | 6.7 | **1.001** |
| Variant call. | 48 | 26.1 | 26.7 | 1,254.0 | 1,283.8 | 4.9 | 2.4 | **1.024** |
| Piper | 98 | 0.6 | 1.0 | 58.5 | 96.5 | 0.6 | 2.6 | **1.650** |

The mean pipeline execution time in the native environment was 1,156.9 min (19 h 16 m 55 s), while the mean execution time when running it with Docker was 1,158.2 min (19 h 18 m 14 s). Thus, the containers execution introduced a virtually zero overhead of 0.1% (see Table 1 and Fig. 1).

## Benchmark 2

The second benchmark was executed using an assembly-based variant calling pipeline, part of a "Minimum Information for Reporting Next Generation Sequencing Genotyping" (MIRING)-compliant genotyping workflow for histocompatibility, immunogenetic and immunogenomic applications (*Mack et al., 2015*).
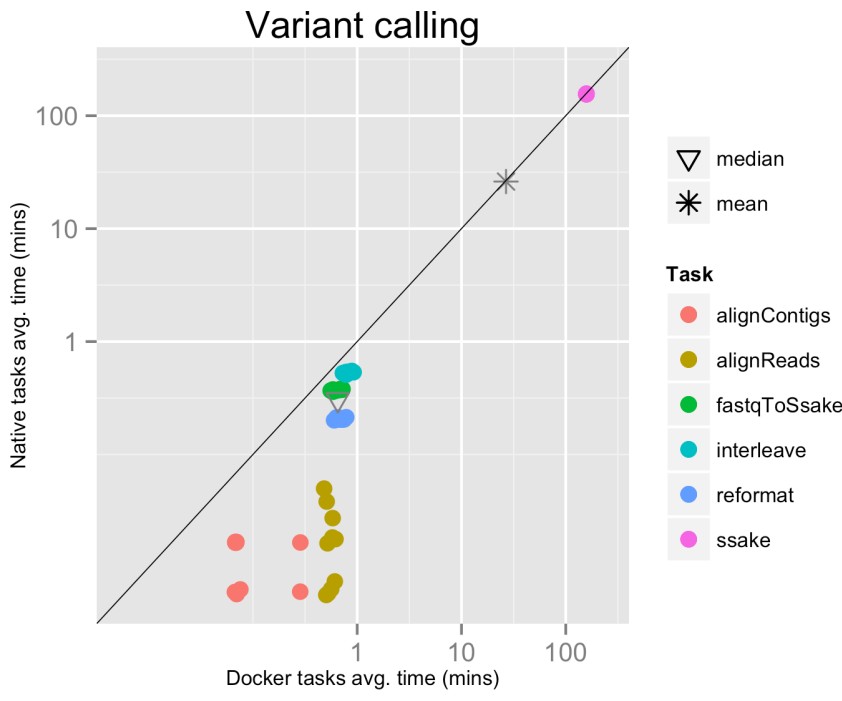

**Figure 2 Variant calling pipeline tasks, native vs. Docker mean execution times.** Time elapsed (in minutes) to complete since the submission including the container instantiation. Each point represents the mean task time for the same type in one pipeline run.

Paired-end genomic reads from targeted human leukocyte antigen (HLA) and killer-cell immunoglobulin-like receptor (KIR) genes are assembled into consensus sequences. Reads and consensus sequences are then aligned to the human genome reference and used to call variants.

The pipeline was launched 10 times using Illumina paired end genomic reads targeted for major histocompatibility complex (MHC) class I HLA-A, HLA-B, and HLA-C genes and MHC class II gene HLA-DRB1 from 8 individuals. The following versions of these tools were used both in the native and Docker environment: ngs-tools 1.7, SSAKE 3.8.2 (*Warren et al., 2007*), BWA 0.7.12-r1039 (*Li & Durbin, 2009*), Samtools 1.2 (*Li et al., 2009*).

Each run executed 48 tasks, and the maximum number of tasks that could be executed in parallel was set to 10. Most of the tasks completed in a few seconds, with the exclusion of the SSAKE stage which needed from 2 to 3.5 h to complete (see Fig. 2).

The mean pipeline execution time in the native environment was 1,254.0 min (20 h 53 m 58 s), while the mean execution time when running it with Docker was 1,283.8 min (21 h 23 m 50 s). This means that when running with Docker the execution was slowed down by 2.4% (see Table 1).

## Benchmark 3

The last benchmark was carried out using Piper-NF, a genomic pipeline for the detection and mapping of long non-coding RNAs.

The pipeline takes as input cDNA transcript sequences in FASTA format, which are aligned using BLAST against a set of genomes also provided in FASTA format. Homologous regions on the target genomes are used as anchor points and the surrounding regions are then extracted and re-aligned with the original query. If the aligner can align these sequences and the alignment covers a required minimal region of the original query, the sequences are used to build a multiple sequence alignment that is then used to obtain the similarity between each homologous sequence and the original query.

As in previous experiments, the pipeline was run 10 times using the same dataset with and without Docker. We used as the input query a set of 100 RNA-Seq transcript sequences in FASTA format from the Gallus gallus species. The input sequences were mapped and aligned towards a set of genomes consisting of *Anas platyrhynchos*, *Anolis carolinensis*, *Chrysemys picta bellii*, *Ficedula albicollis*, *Gallus gallus*, *Meleagris gallopavo*, *Melobsittacus undulatus*, *Pelodiscus sinensis*, *Taeniopygia guttata*, from Ensembl version 73. The following versions of the tools were used both in the native and Docker environment: T-Coffee 10.00.r1613 (*Notredame, Higgins & Heringa, 2000*), NCBI BLAST 2.2.29+ (*Altschul et al., 1990*), Exonerate 2.2.0 (*Slater & Birney, 2005*). Each run executed 98 jobs and the maximum number of tasks that could be executed in parallel was set to 10.

The mean pipeline execution time in the native environment was 58.5 min, while the mean execution time when running it with Docker was 96.5 min. In this experiment, running with Docker introduced a significant slowdown of the pipeline execution time, around 65% (see Table 1).

This result can be explained by the fact that the pipeline executed many short-lived tasks: the mean task execution time was 35.8 s, and the median execution time was 5.5 s (see Fig. 3). Thus, the overhead added by Docker to bootstrap the container environment and mount the host file system became significant when compared to the short task duration.

## RESULTS

In this paper, we have assessed the impact of Docker containers technology on the performance of genomic pipelines, showing that container "virtualization" has a negligible overhead on pipeline performance when it is composed of medium/long running tasks, which is the most common scenario in computational genomic pipelines.

Interestingly for these tasks the observed standard deviation is smaller when running with Docker. This suggests that the execution with containers is more "homogeneous," presumably due to the isolation provided by the container environment.

The performance degradation is more significant for pipelines where most of the tasks have a fine or very fine granularity (a few seconds or milliseconds). In this case, the container instantiation time, though small, cannot be ignored and produces a perceptible loss of performance.

Other factors that don't have a direct impact on the execution performance should be taken in consideration when dealing with containers technology. For example, Docker images, though smaller than an equivalent virtual machine image, need some time in order

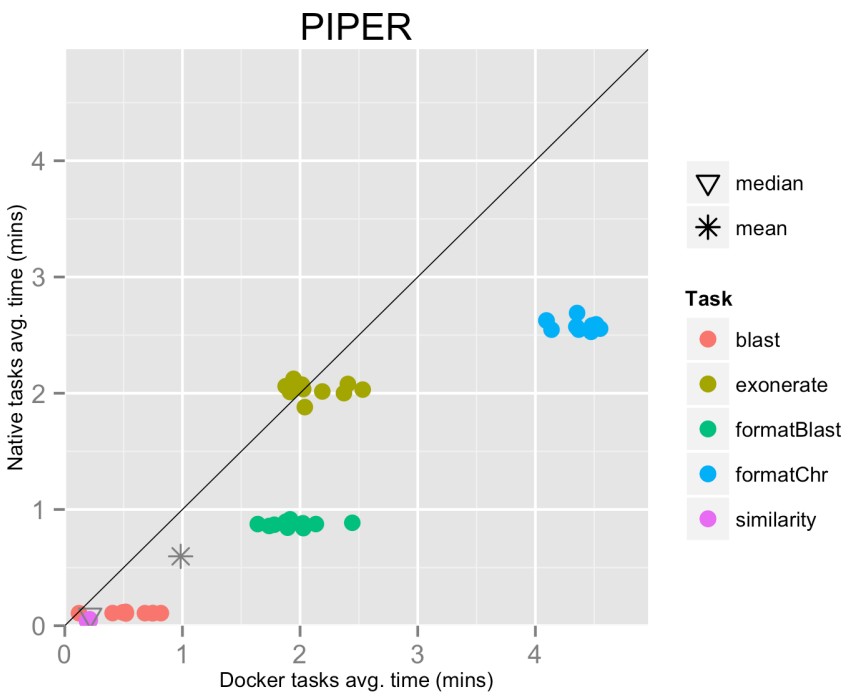

**Figure 3 Piper pipeline tasks, native vs. Docker mean execution times.** Time elapsed (in minutes) to complete since the submission including the container instantiation. Each point represents the mean task time for the same type in one pipeline run.

to be downloaded from a remote repository. This time depends on the available Internet connection and the image size (commonly several hundreds megabytes). Installing a local repository mirror that allows images to be cached in the local organization network is a good strategy to speed-up their download.

Docker images can also be created from scratch using a *Dockerfile* i.e., a simple text file that lists the dependencies and the commands necessary to assemble the target image (e.g., downloading and installing software packages, setting environment variables, etc.). The image-building process is managed automatically by the Docker engine and commonly requires a few minutes. Since a *Dockerfile* is a small resource, it can easily be shared with other researchers. Thus, it can be used as a "cheap" alternative to exchange pre-built Docker image binary files (although this approach isn't as reliable as storing an image in a Docker repository because, due to volatility of software components and web resources, the required dependencies may be changed or no longer available when trying to build the image).

The Docker tool is relatively easy to use as long as you only need to execute a few containers. However in order to "dockerize" the overall execution of a multistage pipeline, where each step may require the execution of a different container, it is not such a straightforward task and can result in unnecessary complexity in the data analysis workflow. In this case, a tool like Nextflow, which handles the execution of Docker containers transparently, can greatly simplify the resulting application.

It is important to stress that Docker is still a young technology with some limitations and potential security problems that should be taken into consideration when dealing with it. For example, despite the fact that Docker takes advantage of the Linux kernel's ability to create isolated environments in which each container receives its own network stack, process space and file system, this separation is not as strong as that of virtual machines which run an independent OS instance on top of hardware level virtualization. As of this writing, Docker does not provide user namespace isolation. This means that a process running in a container, with for example user ID 1000, will have the privileges of that user ID on the underlying system. In the same way, a process running in the container as *root* has root-level privileges on the underlying host when interacting with the kernel. As a consequence, a malicious user can easily gain unrestricted access to the hosting file system by simply mounting the root path in a container running with root permissions.

Finally, it is worth noting that Docker, currently, can only be installed natively only on Linux based operating systems. Available installations on Mac OS-X and Windows include or require a complete Linux virtual machine layer to operate properly. Thus, any performance benefits would be lost.

## CONCLUSION

The fast start-up time of Docker containers allows one to virtualize a single process or the execution of a bunch of applications, instead of a complete operating system. This opens up new possibilities such as the ability to chain together the execution of multiple containers as it is commonly done with computational workflows, or the possibility to "virtualize" distributed job executions in an HPC cluster of computers.

In this paper, we show that Docker containerization has a negligible impact on the execution performance of common genomic pipelines where tasks are generally very time consuming.

The minimal performance loss introduced by the Docker engine is offset by the advantages of running an analysis in a self-contained and precisely controlled runtime environment. Docker makes it easy to precisely prototype an environment, maintain all its variations over time and rapidly reproduce any former configuration one may need to re-use. These capacities guarantee consistent results over time and across different computing platforms.

Though this work takes in consideration a limited but representative subset of bioinformatics tools and data analysis workflows, we expect that our findings can be generalized to other computational analysis having similar resource requirements and task granularity compositions.

## ACKNOWLEDGEMENT

We acknowledge the support of the Spanish Ministry of Economy and Competitiveness, "Centro de Excelencia Severo Ochoa 2013-2017," SEV-2012-0208.

### Funding

The authors received no funding for this work.

### Competing Interests

The authors declare there are no competing interests.

### Author Contributions

- Paolo Di Tommaso conceived and designed the experiments, performed the experiments, analyzed the data, contributed reagents/materials/analysis tools, wrote the paper, prepared figures and/or tables, reviewed drafts of the paper.
- Emilio Palumbo, Maria Chatzou and Michael L. Heuer contributed reagents/materials/analysis tools, wrote the paper, reviewed drafts of the paper.
- Pablo Prieto analyzed the data, contributed reagents/materials/analysis tools, wrote the paper, prepared figures and/or tables, reviewed drafts of the paper.
- Cedric Notredame conceived and designed the experiments, contributed reagents/materials/analysis tools, wrote the paper, reviewed drafts of the paper.

### Data Availability

Github: https://github.com/cbcrg/docker-benchmarks.

### Supplemental Information

Supplemental information for this article can be found online at http://dx.doi.org/10.7717/peerj.1273#supplemental-information.

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
