# Peer review of "The impact of Docker containers on the performance of genomic pipelines"

_PeerJ, doi:10.7717/peerj.1273_

## Round 0.1 · original submission · Minor Revisions

The three reviewers agreed that the manuscript will be of interest to researchers interested in developing reproducible pipelines. However, as noted by both reviewers 1 and 2, the submitted manuscript does not provide sufficient information to reproduce the study (e.g., the pipeline scripts need to be available to the readers) or to fully understand what data are reported (see reviewer #2, comments: how are the measurements performed exactly, and what type of execution time is reported).

After reading the manuscript, I also agree with Reviewer #2 that the manuscript would be strengthened by providing a more balanced description of the limitations of docker container technology, which go beyond performance. An indication of how long it took the authors to build the images necessary for this evaluation would be useful, as well as a discussion of non-performance criteria.

I believe that the manuscript will be strengthened if most comments of the reviewers were addressed and if statistics of variation were provided for performance data (e.g., reporting median execution time without discussion of variability of execution times is less than optimal, unless the variations are too small to matter. Are the variances similar with container technology and could variance of the measurements explain the "unintuitive" observation pointed out by reviewer #2?).

·

Basic reporting

The authors provide a concise and careful demonstration that the containerization of all individual elements of a bioinformatics pipeline has only negligible impact on the computational performance of that pipeline. As such, this work clearly demonstrates that the containerized approach implemented by Docker has solved a long-standing barrier to virtualization in bioinformatics, in which it rapidly becomes infeasible to chain together large numbers of virtual machines.

The article is clearly written with appropriate introduction, context, and citations, and in a style and format consistent with PeerJ policies. Figures are relevant, clear, and appropriately labeled. The authors might consider some more explicit citations to previous suggestions regarding the use of more heavy-weight virtual machines for bioinformatics pipelines, along with some discussion which would better place the concern of performance in context, e.g. see doi:10.1126/science.325_1622b vs doi:10.1126/science.1174225, also doi:10.1109/MCSE.2012.62, doi:10.1038/nbt1110-1181; though my favorite citation for this is: https://twitter.com/BioMickWatson/status/265037994526928896

Experimental design

While raw data is displayed visually, a plain-text file (e.g. csv file) of the data shown would probably be the preferable way to comply with PeerJ data sharing requirements. More importantly however, I would recommend that the authors provide the scripts sufficient to replicate the computational analysis shown here (as required by PeerJ policies), or at very least provide links to the Dockerfiles involved. The primary weakness of the manuscript is the lack of sufficient documentation as to precisely how the containers were executed in sequence and how the results of one step in the pipeline were passed to the next. This must be clarified in both the text and provided as supplement (or links).

Validity of the findings

The results generally appear well-supported by the data provided, though of course curious minds might wonder about other circumstances from the test cases considered here, such as different hardware configurations. These may safely be considered beyond the scope of the analysis shown here. One issue does stand out as needing further discussion at least:

The authors note (lines 61-65) that:

> It should be noted that when the pipeline is executed with Docker support it does not mean that the
> overall pipeline execution runs "inside" a single container, but that each task spawned by the pipeline
> runs in its own container. This approach allows a Docker based pipeline to use a different image for each
> different task in the computational workflow, and therefore scale seamlessly in a cluster of computers
> (which wouldn't be possible using the single container approach).

this is perhaps the most interesting part of the paper, and deserves to be expanded. In particular, there are several different mechanisms to pass data from one container to the next, each of which may have very different performance implications. The description provided here is insufficient to deduce the approach used. Notably, not all approaches that use different containers for different tasks can be scaled 'seamlessly in a cluster' as the authors suggest. Perhaps the simplest mechanism is though linking local volumes, which does not necessarily scale to a multiple-node approach (unless the cluster has been configured with shared disk space, such as NFS). Other approaches, such writing data to dedicated database containers (e.g. MySQL, postgresql, redis) can avoid this but may introduce more significant overhead. Notably, the authors note that all tests were run on a single node of a cluster, where these issues do not arise. (Of course given that the experimental design requires this, such that the results may be directly compared to the same software running without containerization, the fact that they use a single node is a feature and not a bug of this research. Nevertheless, since they comment on the additional non-negligible advantage of being able to use a cluster, the point deserves further discussion.

The other important issue the authors should note is that the popular Docker installations on Mac and PC machines currently do require complete virtualization to provide Docker on OS's that lack the Linux kernel. As such, performance would be lost, though not do to Docker directly but rather the intermediate virtualization layer created by Kitematic or boot2docker (e.g. Virtualbox). The authors should simply flag this issue so as not to confuse readers on these operating systems.

·

Basic reporting

* Saying that Docker containers are an "ideal solution" is stating it too strongly. Any solution will have strengths and limitations. The authors should be more explicit about why Docker is helpful in addressing the issues raised.

* Need reference(s) for this statement: "In the past virtual machines were proposed as an answer to this issue."

* "thus raising serious reproducibility issues" -> Please be more explicit about what reproducibility means and why it is a concern.

* Please add some kind of reference to Docker (maybe to their web site?).

* There are many grammar problems that should be addressed to make the manuscript more readable.

* There are no figure captions.

* Please define what "KVM virtualization" is. Also, what is the relationship between software containers and KVM virtualization? Also explain briefly why the IBM study was insufficient to illustrate the Docker containers have a minimal performance penalty for genomic analyses.

* The authors mention "the ENCODE analysis." Some readers will not know what this is. Also, there have been multiple ENCODE analyses. Please be more explicit.

* Please be more explicit in this statement: "They are indeed very convenient but come along with a few major issues that include high latency and significant overhead."

* "Docker containers technology has been designed to address these issues." This statement makes it sound like Docker was designed to address this problem in a genomics/research/reproducibility context. Please clarify.

* Please be more explicit in what the following statement means: "Furthermore their experimental nature can result in frequent updates, thus raising serious reproducibility issues."

* "The first performance evaluation was carried out using a simple pipeline for RNA-Seq data analysis (15)." -> This appears to be a numbered citation, but the references are not numbered.

Experimental design

* Can other scientists access the NextFlow workflows that were used for this analysis so they can validate or build upon this analysis?

* At what exact point does the timer start and stop? Does this include time to load the Docker software and containers?

* What method/utility was used to measure execution time? Did it measure "real" time, system time, user time?

Validity of the findings

* It is unintuitive that Docker executed slightly faster for benchmark 1 (RNA-Seq). It would be helpful if the authors could investigate this further to make sure nothing is wrong and offer an explanation on why this might have occurred.

* The authors have demonstrated that in some cases, there is a minimal execution overhead. However, the authors could also mention the overhead of time to create and maintain Docker images and to learn this new technology. Or if the authors feel that these costs are negligible, it would be helpful to state why.

* Along these lines, the authors would do well to address non-performance issues (to some extent). Does using containers introduce any limitations that scientists would not otherwise face? For example, with security? Ability to deploy in an environment where the user does not have root access, etc? I don't think it's necessary to go into exquisite detail here. But it would be valuable to mention that performance is not the only consideration.

* On Table 1 (or in a figure), it would be helpful if the authors indicated how much variation there was across the pipeline runs, so we can get a sense for how consistently these patterns are observed.

* It would be interesting if the authors could speculate (in the Discussion) about whether these findings might generalize to other computationally intensive research areas (not just genomics).

Additional comments

The authors address an important issue that computational scientists currently face: will there be a considerable performance penalty in using software containers for research? The authors have addressed this topic adequately. However, I feel that various issues should be addressed to improve the quality of the manuscript. I have outlined these below. I make the disclaimer that I am not an operating-systems person, so my ability is limited to assess the more technical aspects of containerization.

Reviewer 3 ·

Basic reporting

"No Comments"

Experimental design

"No Comments"

Validity of the findings

"No Comments"

Additional comments

Docker containers (an OS level virtualization for Linux) solve the problem of easily distributing reproducible software. In particular genomic pipelines often consist of various third party tools and thus can profit from Docker. The novel contribution of the authors in this paper is a description of the overhead introduced by Docker containers in context of genomic pipelines and the different scenarios where this overhead can be problematic or is negligible.

The authors evaluate the performance of Docker using three representative genomic pipelines (using the Nextflow tool), a RNA-Seq data analysis pipeline, a variant calling pipeline, and a pipeline for detection and mapping of long non-coding RNAs.

The authors show that with exception of very short running tasks, the overhead of Docker containers is negligible.


Major
- Related work such as the Skyport workflow system that also uses Docker containers should be referenced: Gerlach et al, DataCloud 2014.

Minor
- first sentence of abstract: grammar is confusing/wrong ?
- commas missing: Line 21: "Furthermore, their..." , 22: "In the past, ...", 53: "moreover,"

---

## Round 0.2 · accepted · Accept

Thank you for revising the manuscript and carefully addressing the comments of the reviewers. I appreciate that you provided clear supplementary material to complement the method descriptions and make it possible to understand precisely how the comparisons were performed.